# Real-time tracking of coherent oscillations of electrons in a nanodevice by photo-assisted tunnelling

Yang Luo [1], Frank Neubrech [1,2], Alberto Martin-Jimenez[1], Na Liu [1,2], Klaus Kern [1,3] & Manish Garg [1] ✉

Coherent collective oscillations of electrons excited in metallic nanostructures (localized surface plasmons) can confine incident light to atomic scales and enable strong light-matter interactions, which depend nonlinearly on the local field. Direct sampling of such collective electron oscillations in real-time is crucial to performing petahertz scale optical modulation, control, and readout in a quantum nanodevice. Here, we demonstrate real-time tracking of collective electron oscillations in an Au bowtie nanoantenna, by recording photo-assisted tunnelling currents generated by such oscillations in this quantum nanodevice. The collective electron oscillations show a noninstantaneous response to the driving laser fields with a $T_2$ decay time of nearly 8 femtoseconds. The contributions of linear and nonlinear electron oscillations in the generated tunnelling currents were precisely determined. A phase control of electron oscillations in the nanodevice is illustrated. Functioning in ambient conditions, the excitation, phase control, and read-out of coherent electron oscillations pave the way toward on-chip light-wave electronics in quantum nanodevices.

Interaction of light with metallic nanostructures can lead to a collective oscillation of conduction electrons. If the frequency of the incident light matches with the intrinsic resonance frequency of the collective electron oscillations (surface plasmons) in nanostructures, such oscillations can be dramatically amplified[1,2]. The resulting strong electromagnetic field arising from the driven collective electron oscillations has now found many applications, ranging from atomic scale nano-optics[3–6] to single molecule sensing[7]. Moreover, it enables exploring the nonlinear optical response of matter[6,8–19]. On interaction of strong electromagnetic fields with matter, nonlinearity in electron oscillations sets in, implying that the electron motion does not remain harmonic anymore. Anharmonic electronic motion implies that electrons oscillate with many frequencies, which are multiples of the driving frequency, in close analogy to a classical anharmonic (driven) oscillator.

In absence of the capability to directly resolve coherent electron oscillations in the time domain, interaction of light with matter has been studied by spectral measurements (in the UV to infrared range) utilizing the techniques of absorption spectroscopy and transient reflectivity[20,21]. Signatures of nonlinearity in light-matter interactions in nanostructures have also been studied by spectral measurements, e.g. second and third harmonic generation[22–24]. Ultrafast techniques such as time-resolved two-photon photoemission[25,26] (TR-2PPE) and time-resolved scanning near-field optical microscopy[27,28] (TR-SNOM) have been successfully applied to monitor ultrafast plasmon dynamics at the nanoscale. Such measurements can retrieve the ultrafast evolution of the spatially dependent plasmonic electric fields[29,30], nevertheless, they do not capture the phase information of the frequency-dependent plasmon oscillations. Recent experiments have shown that the photo-assisted tunnelling currents induced in a plasmonic nanodevice can be controlled by tuning the carrier-envelope-phase (CEP) of the driving laser pulses[15,19]. Nevertheless, a direct sampling of the ultrafast coherent collective electron oscillations and the resulting local electric field in the time domain has not been reported yet, which

[1]Max Planck Institute for Solid State Research, Heisenbergstr. 1, 70569 Stuttgart, Germany. [2]2nd Physics Institute, University of Stuttgart, Pfaffenwaldring 57, 70569 Stuttgart, Germany. [3]Institut de Physique, Ecole Polytechnique Fédérale de Lausanne, 1015 Lausanne, Switzerland. ✉e-mail: mgarg@fkf.mpg.de

is highly desired since it is the key to modern photonic functionalities operating at petahertz frequencies, ultrafast switching, and all-optical signal processing[31–35].

Here, we demonstrate an approach wherein both plasmon oscillations and nonlinear electron oscillations arising from the nonlinear optical response induced by ultrashort laser pulses in a strongly light-interacting quantum nanodevice can be traced directly in the time domain. Our nanodevice comprises of Au bowtie nanoantennas, with junction gaps of only a few nm. Electron oscillations in the nanodevice were traced by recording photo-assisted tunnelling currents. The plasmon oscillations show a noninstantaneous response to the driving electric field of the laser pulses with a $T_2$ decay time of ~8 fs. The spectral phase of the plasmonic field is highly dispersive in close agreement with a classical harmonic oscillator driven at its resonance frequency. Furthermore, we show that the contributions of linear and nonlinear electron oscillations in the generated photo-assisted tunnelling currents can be precisely deciphered and sampled in real-time.

Phase control of plasmon oscillations directly in the time domain in the quantum nanodevice is also demonstrated.

## Results

### Real-time sampling of coherent collective electron oscillations

In our experiments, arrays of seven identically designed Au bowtie nanoantennas (Fig. 1a) of ~300 nm size (isosceles triangle) with junction gaps of a few nm fabricated on top of a fused silica substrate (see Methods for details) were illuminated with two ultrashort laser pulses (pulse duration, $\tau_P$ ~ 7 fs) of slightly different carrier frequencies. Fig. 1b shows a scanning electron microscope (SEM) image of seven Au bowtie nanoantennas. The plasmonic response of these nanoantennas as a function of the incident laser wavelength and the spatial distribution of the local field enhancement in the junction were calculated by finite element simulations, as shown in Fig. 1c (see section II in Supplementary Information for details). Owing to the high field enhancement at the junctions of the bowtie nanoantennas, plasmon

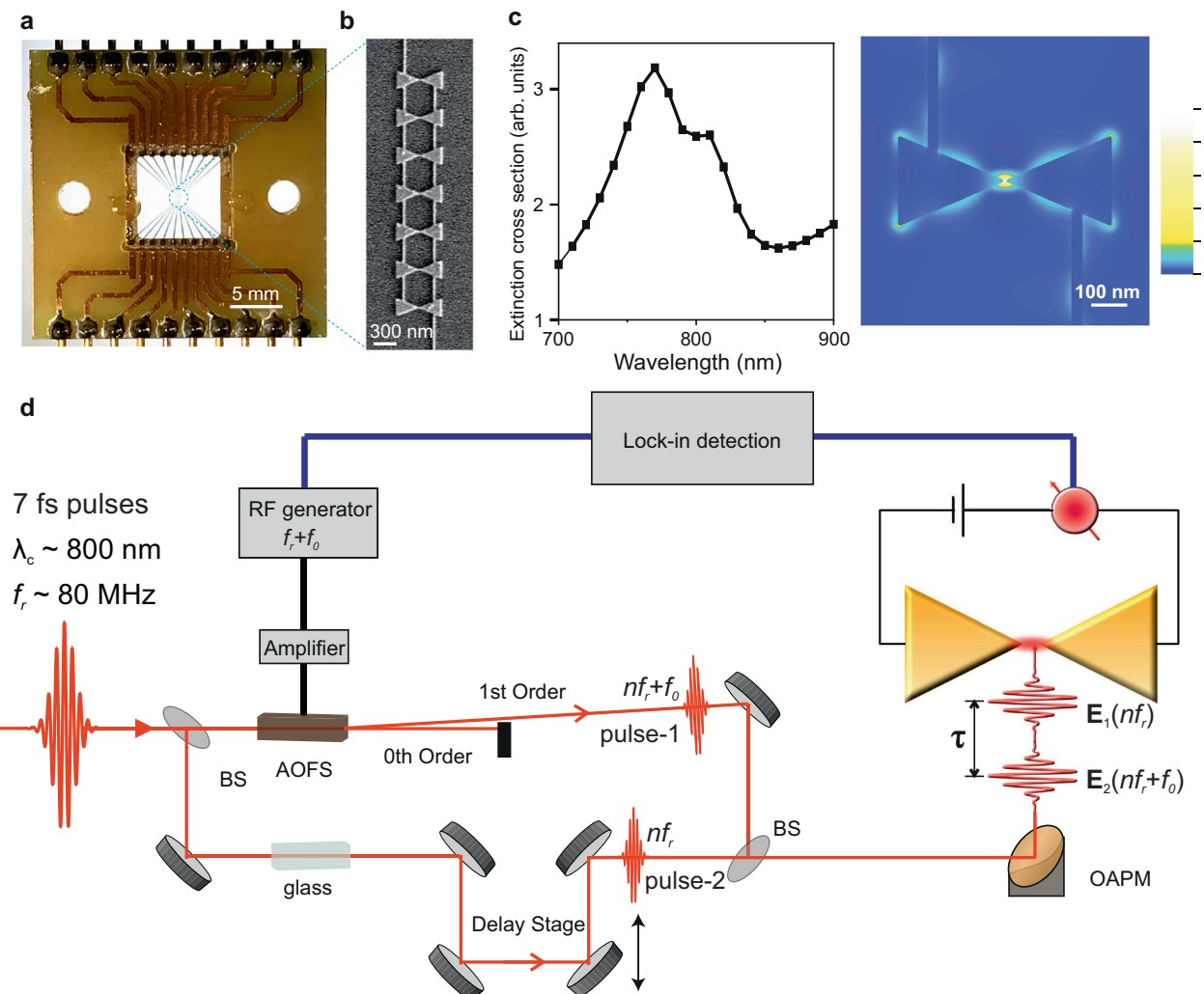

**Fig. 1 | Nanodevice and optical homodyne beating technique for tracing coherent oscillations of electrons. a** Photograph of the nanodevice. The nanodevice consists of a series of seven connected, identically designed, Au bowtie nanoantennas fabricated on top of a fused silica substrate. **b** A scanning electron microscope (SEM) image of seven bowties. **c** Numerically calculated plasmonic response (left panel) of a single Au bowtie (junction size of 10 nm) showing the 2nd order resonance as deduced from the spatial near field distribution (right panel) at 770 nm. The local field-enhancement factor is denoted by the colour code in the colour bar. In the simulations, the electrical field is polarized along the long bowtie axis. **d** Schematic of the experimental set-up: laser pulses with a very small offset frequency, $f_0$, in their carrier frequencies are generated by selecting the orginal laser beam $\mathbf{E}_1(nf_r)$ (pulse-2) and first-order diffracted beam $\mathbf{E}_2(nf_r + f_0)$ (pulse-1) of laser pulses (pulse duration $\tau_P$ ~ 7 fs, central wavelength $\lambda_c$ ~ 800 nm, repetition rate $f_r$ ~ 80 MHz) traversing through an acousto-optic frequency shifter (AOFS). Those two pulses are delayed ($\tau$) and combined, then focused onto the nanodevice. Photocurrent generated by the laser pulses in the nanodevice is measured by lock-in detection at the offset frequency of $f_0$. BS Beam splitter. AOFS Acousto Optic Frequency Shifter. RF generator Radio frequency signal generator. OAPM Off-Axis Parabolic Mirror.

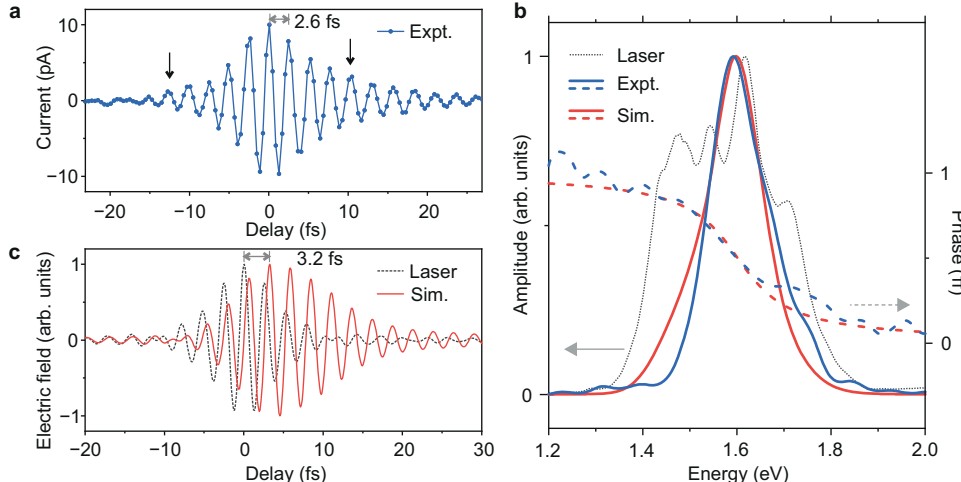

**Fig. 2 | Real-time tracking of coherent collective electron oscillations.**
**a** Variation of the laser-induced photocurrent as a function of the delay between pulse-1 and pulse-2 laser pulses of slightly different carrier frequencies (Fig. 1d) in a biased nanodevice, with the bias in the nanoantenna junction being 2.5 V. The designed gap size of the nanoantennas was 5 nm. **b** Comparison of the experimental and calculated plasmonic response in the nanoantenna junction. The dashed-blue and dashed-red curves represent the phases of experimentally measured and theoretically simulated plasmon oscillations, respectively. The dotted-black curve shows the spectrum of the incident laser pulses on the nanoantenna junction. **c** Comparison of the electric field of ~ 7 fs long driving laser pulse (dashed black curve) with the theoretically calculated plasmonic field (solid blue curve). Gray double arrow indicates a delayed response of the plasmon oscillations with respect to the electric field of the driving laser pulse.

oscillations will dominantly be excited when focusing the incident laser pulses in these nanoantenna junctions. In order to time resolve the plasmon oscillations induced by the ultrashort laser pulses, we probe the homodyne beating signal between plasmon oscillations induced by two ultrashort laser pulses with a very small difference in their carrier frequencies.

Here, we briefly explain the technique of homodyne beating, a self-referencing method to measure e.g. the phase of plasmon oscillations that are induced by the laser pulses in the nanoantennas (see section III in Supplementary Information for details). Ultrashort laser pulses ($\tau_P$ ~ 7 fs) coming at a repetition rate of ~80 MHz ($f_r$) were passed through an acousto-optic-frequency-shifter (AOFS), which is driven at the frequency of $f_r + f_O$ as shown in Fig. 1d, $f_O$ is ~700 Hz. The 1st order diffraction beam ('pulse-1') coming from the AOFS is slightly shifted in its carrier frequency, by $f_O$, with respect to the carrier frequency of the incident (original) laser pulses ('pulse-2'). These two laser pulses are then combined and focused in the junction of the nanoantennas, as schematically shown in Fig. 1d. The excited collective electron oscillations in the junction produce a local electric field, which can be expressed as a convolution of the incident laser field $E_i(t)$ and the optical response function ($R(t)$) of the nanoantennas; $E_{Li}(t) \propto E_i(t) * R(t)$, where the subscripts, $i = 1, 2$ denote the two different laser pulses. The net electric field produced by the electron oscillations induced by the two laser pulses at the junction of nanoantennas can then be expressed as;

$$E(t) = \sum_n \{E_{L1}(t) \exp(-inf_r t) + E_{L2}(t-\tau) \exp(-i(nf_r + f_0)(t-\tau)) + c.c.\}$$

(1)

where $nf_r$ is the nth (range from ~3.5 $\times 10^6$ to ~6 $\times 10^6$) multiple of the repetition rate of the laser pulses, $\tau$ represents the delay between the two pulses.

Due to the highly localized field enhancement in the nanoantenna junction, only the electrons close to the junction can be excited on interaction with photons and tunnel across the junction. A bias voltage is applied on the anoantenna to facilitate the electron tunnelling processes. The mechanism of photo-assisted tunneling in the nanoantenna junction will be discussed later in the text. The single-photon assisted tunnelling of electrons induced e.g. by the plasmon oscillations in the nanoantenna junction is proportional to the linear polarization of the system; $I_1^T(t) \propto E(t)^2$, which contains terms oscillating at multiple frequencies. The very high-frequency terms, i.e. twice the carrier frequencies of the laser pulse, $2nf_r$ and $2nf_r + 2f_O$ (~0.6 $\times 10^{15}$ Hz), cannot be lock-in detected. Nevertheless, an interference term arising due to interference of the plasmon oscillations induced by the two laser pulses comes at the very small offset frequency of $f_O$ between the laser pulses; $I_1^T(t) \propto E_L(t)^2 \cos(f_0 t + nf_r \tau)$, assuming $E_L(t) = E_{L1}(t) = E_{L2}(t)$. This interference term contains both the amplitude and the phase information of the local electric field in the nanoantenna junction, which allows for a direct characterization of the plasmon oscillations.

A real-time sampling of the plasmon oscillations induced by the laser pulses with a total pulse energy of ~100 pJ is shown in Fig. 2a. The plasmons undergo an oscillation period of ~2.6 fs. Fourier transform of the time trace in Fig. 2a reveals the spectral shape of the plasmonic response of the nanoantennas (Fig. 2b), which closely resembles the spectral shape of the plasmonic response evaluated from the finite element simulations (Fig. 1c and Fig. S2 of the Supplementary Information). A comparison of the spectrum of this plasmonic response with the spectrum of the incident laser pulses reveals significant spectral and temporal shaping of the laser pulses in the nanoantenna junction, as shown in Fig. 2b. A broad plasmonic response of the nanoantennas (Fig. 1c) implies a very fast damping rate of the induced plasmon oscillations, in the range of only a few fs[36]. The decay of the plasmon oscillations can be seen by a long and asymmetric oscillation trace on the positive side of the delay axis in Fig. 2a, as indicated by the black arrows.

Non-resonant excitation of bound electrons in a system is usually instantaneous, i.e. the electrons will oscillate in phase with the driving electric field and the oscillations will fade out as soon as the impetus from the driving field is over[37]. However, when the electrons are excited on resonance, their response is delayed with respect to the driving electric field and it has a much longer decay time. Such delayed response of bound electrons has been earlier reported for a dielectric medium[37,38] as well as for an atom[39]. Furthermore, another key distinctive feature of the resonant electron oscillations compared to the non-resonant case is their phase curve along the frequency axis. The

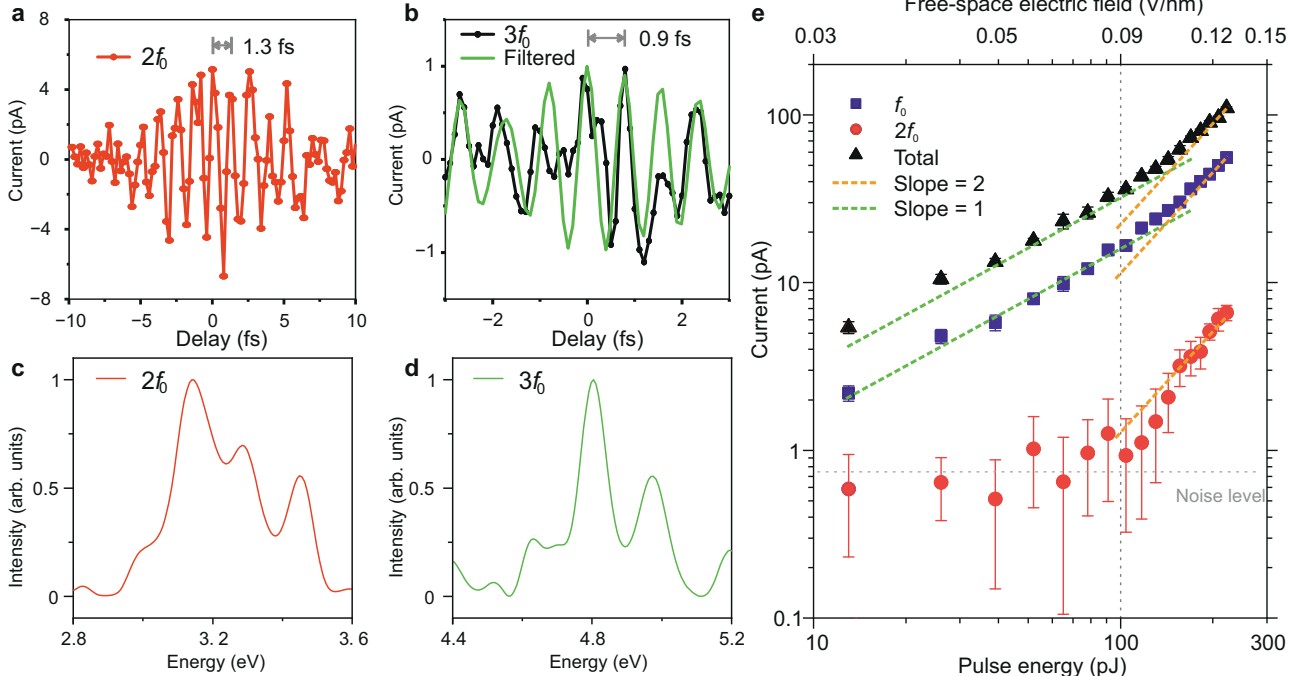

**Fig. 3 | Quantifying single and multi-photon light-matter interactions.**
**a**, **b** Variation of the laser-induced photocurrent as a function of the delay between pulse-1 and pulse-2 laser pulses measured at the lock-in frequency of $2f_0$ and $3f_0$, respectively. The bias in the nanoantenna junction is 3.0 V. **c** Spectral response of the time-resolved electron oscillations in **a** (red-curve). **d** Spectral response of the time-resolved electron oscillations in (**b**) (green-curve). **e** Measured variation of the photo-assisted tunnelling current as a function of increasing field-strength of the incident laser pulses (top x-axis) on the nanodevices. Violet and red-points show the variation of the photocurrent signal measured with the lock-in detection frequency of $f_0$ and $2f_0$, respectively. Measurements were performed at the zero-delay between pulse-1 and pulse-2 (Fig. 1d). Peak-field strength refers to the maximum of the net electric field produced by the two pulses. Black-points show the variation of the total photo-assisted tunnelling current generated in the nanodevice. Dashed green and orange curves indicate a slope of one (linear) and two (quadratic) in the dual logarithmic plot. Error bars represent the standard deviation of the signal from ten consecutive measurements.

phase curve around the resonance frequency is very dispersive, and the phase difference along the two extrema of the resonance frequency is ~$\pi$ radians.

A bound electronic system, such as plasmons in the nanoantenna junction with the restoring force from the ions of the nanoantennas, can be simply modelled as a driven damped harmonic oscillator[40,41]. The electric field of the plasmon oscillations ($E_{Pl}$) can be expressed as;

$$E_{Pl}(t) \propto \int_{-\infty}^{t} \frac{1}{\omega_R} E_{Laser}(t') \exp(-\gamma(t-t')) \sin(\omega_R(t-t')) dt' \quad (2)$$

where $\omega_R$ is the resonance frequency of the plasmons, $E_{Laser}$ is the electric field of the driving laser pulse, and $\gamma$ is the intrinsic damping rate of the plasmon oscillations, which is determined by the bandwidth of the spectrum of the plasmonic resonance.

Fig. 2c shows a comparison of the calculated plasmonic field, considering a resonance frequency $\omega_R$ at ~1.6 eV and a damping rate $\gamma$ of ~0.12 fs$^{-1}$, with the incident electric field of ~7 fs Fourier-limited laser pulse. This damping rate corresponds to a spectral linewidth of ~160 meV of the plasmonic response, which is in agreement with the calculations using finite element simulations. The local electric field resulting from plasmon oscillations as measured in the experiment is in good agreement with the calculated plasmonic field as shown in Fig. 2c. A long tail associated with the damping of the plasmon oscillations along the positive delay axis can be clearly seen, which decays on the time scale of ~8 fs ($T_2$ decay time). A direct comparison of the experimentally measured plasmon oscillations, the electric field of the driving laser pulse, and the calculated plasmonic field is shown in Fig. S3 of the Supplementary Information. The simulation also shows that the peak electric field of the plasmon

oscillations is delayed by ~3 fs with respect to that of the driving laser pulse (Fig. 2c).

The spectrum and the phase of the plasmon oscillations as captured in the experiment are contrasted with those obtained from calculations (obtained by Fourier transform of $E_{Pl}$) in Fig. 2b. Plasmons undergo a spectral phase shift of nearly $\pi$ radians across its resonance curve as also consistent with the model. The spectrum of plasmon oscillations as simulated from the model is also in good agreement with the measurement. This consistency further attests the validity of the simple damped harmonic oscillator model describing the plasmon oscillations as measured in the experiments, and transparently demonstrates the capability of our technique to time resolve ultrafast plasmon oscillations. In addition to allowing access into the near-field plasmon oscillations ($E_{Pl}$), an inversion of Eq. (2) also enables a direct measurement of the far field of the driving laser pulse.

### Unravelling linear and nonlinear contributions in light-matter interaction

At higher field strength of the incident laser pulses, nonlinear electron oscillations induced by higher order polarization responses of the nanoantenna junction set in due to the strong plasmon-enhanced light-matter interaction[42] (see section III in the Supplementary Information). Fig. 3a shows the temporal evolution of the local second-order nonlinear oscillation of electrons induced at a total pulse energy of ~200 pJ. The photocurrent generated in the nanodevice due to the second-order nonlinear polarization response of the nanoantennas can also be measured with the homodyne beating technique; $I_2^T(t) \propto E(t)^2 E(t-\tau)^2 (1 - \cos(2f_0 t + 2nf_r \tau))$. A lock-in detection of the photocurrent signal at twice the offset frequency ($2f_0$) enables temporal sampling of the second-order nonlinear electron oscillations in the junction; entailing its complete phase

information. At higher pulse energy, ~300 pJ, the third-order nonlinear oscillations of electrons, induced by three-photon absorption, can be recorded as shown in Fig. 3b; measured by performing lock-in detection at $3f_O$ frequency. However, the measurements at such high laser pulse energies are not very stable as the nanodevice is prone to physical damage, which can be induced by electromigration or field evaporation[19]. The oscillation periods of nonlinear electron oscillations for the case of 2nd and 3rd order optical responses are ~1.3 fs and ~0.9 fs, respectively. The spectral responses of 2nd and 3rd order nonlinear electron oscillations reveal a significant spectral shaping due to the multi-photon interactions in the nanoantenna junction, as shown in Figs. 3c, d.

In the weak-field or perturbative regime, light-matter interaction is usually characterized by a power-scaling experiment, wherein a physically relevant quantity is measured as a function of increasing intensity of the laser pulses. A nth order nonlinearity implies the interaction with n number of photons to be dominated[43,44]. Nevertheless, n − 1 and n − 2 photon orders in the light-matter interaction do not cease to exist, but their contributions are harder to access. A technique capable of deciphering the contribution of all photon-channels (linear and nonlinear polarizations) at a particular intensity of the laser pulse in the process of light-matter interaction is considerably sought after. Here, we demonstrate the technique of homodyne beating as a powerful tool to precisely decipher the contributions of different photon channels in the light-matter interaction.

The variation of the total photocurrent generated by the laser pulses in the junction of the nanoantennas as a function of increasing intensity is shown in Fig. 3e (black data points), measured by intensity modulation (at ~520 Hz) of laser pulses. In a dual logarithmic plot, the scaling of the total photocurrent shows a switch from a linear scaling (slope of 1) at lower intensities to a quadrating scaling (slope of 2) at higher intensities of the laser pulses, indicating the contributions from both linear and nonlinear polarization responses of the nanoantenna junction at higher intensities. In order to disentangle the contributions of the different photon-channels, the variation of the photocurrent signal at zero delay between pulse-1 and pulse-2 (Fig. 1d) was measured as a function of intensity of the laser pulses for two different frequencies in the lock-in detection, $f_O$ and $2f_O$, as shown in Fig. 3e. The scaling of the lock-in signal at $f_O$ frequency is similar to the behaviour of the total photocurrent, since this signal can arise from both linear as well as local nonlinear polarization responses of the nanoantenna junction (with a prefactor of 0.5, see section III in Supplementary Information for details). However, the signal at $2f_O$ frequency can only arise from the second-order nonlinear polarization response (with a prefactor of 1/8). Thus, its scaling with respect to the intensity of the laser pulse is purely quadratic. This change from linear (dashed green curve) to quadratic (dashed orange curve) behaviour in the scaling of the total photocurrent signal occurs at the similar local field strength of the laser pulses where the signal at $2f_O$ frequency starts emerging, as indicated by a vertical black-dashed curve in Fig. 3e. Thus, demonstrating a direct measurement of the contribution of the 2nd order light-induced polarization response (nonlinear electron oscillations). We note that the pulse energies were kept below ~220 pJ in the measurement to avoid damage of the nanoantennas. The contribution of photocurrent at $3f_O$ frequency is too weak to be reproducibly determined, but in principle, can be measured with our technique. The local structures in the nanoantennas are unaffected during the intensity dependence measurement (Fig. 3e) as confirmed by the reproducibility-check measurements (see Fig. S6 in Supplementary Information). In conclusion, by recording the homodyne beating signal at $f_O$ and its harmonic frequencies ($2f_O$ and higher), we can precisely determine the contributions of linear and nonlinear electron oscillations (polarization responses) in the generated tunnelling currents in the nanodevice.

## Photo-assisted electron tunnelling in the nanoantenna junction

In the following, we discuss the mechanism of photocurrent generation in the junction of the Au bowtie nanoantennas. The determination of the contributions of one- and two- photon processes in the power scaling measurements of the photocurrent signal in Fig. 3e excludes the mechanism of laser field-driven tunnelling in the nanoantenna junction, where a much less nonlinear power-dependent behaviour is expected. Besides, the Keldysh parameter for our pulses at the nanoantenna junction is above 4, where the laser field-driven tunnelling effects are virtually absent[44]. In order to understand the mechanism underlying the photocurrent generation, we measured the variation of the photocurrent signal at $f_O$ frequency as a function of the increasing bias voltage applied in the nanoantenna junction, as shown in Fig. 4a. The photocurrent signal shows an extremely nonlinear dependence on the applied bias in the junction. Therefore, over-the-barrier photoemission[44] can be excluded, as it would be virtually insensitive to the small biases applied in the junction. Moreover, the electrons excited by plasmon oscillations, following photoexcitation by laser pulses, can be up to approximately 1.6 eV above the Fermi level of Au, but, still significantly below the tunnelling barrier of Au (~5 eV) and cannot induce photoemission (photocurrent) in the measurement.

Here, we describe photocurrent generation by a simple model accounting for tunnelling of electrons across the nanoantenna junction with an effective Fermi electron distribution[45] that is photo-excited by the laser pulses, as shown schematically in Fig. 4b. In the case of photo-assisted tunnelling[46], electrons from one side of the junction are photo-excited via one-, two- or three-photon absorption and then tunnel to the other side through a reduced effective tunnelling barrier. The calculated electron tunnelling probability as a function of increasing bias (see section V in Supplementary Information) for a junction of gap-width of ~1.2 nm matches quite well with the experimentally measured nonlinearity of the photocurrent signal (Fig. 4a). The junction (tunnel) gap of ~1.2 nm is significantly bigger for any DC tunnelling (below 5 V) but not for photo-assisted tunnelling. Using scanning electron microscopy we found that the fabricated junction gap of a single bowtie can be as small as 8 nm. Smaller gap sizes may occur but could not be resolved with the available SEM. The deviations between the designed and the actual junction gap sizes of the nanodevice can result from the exposure characteristics of the resist during the nanofabrications, or the electromigration of Au atoms in the nanoantenna junction[19]. It is worth mentioning that the measurements performed with single bowtie nanoantenna devices (Fig. S5) give similar results as compared to the series of seven identical bowtie nanoantennas (Figs. 2 and 4).

## Real-time phase control of localized plasmon oscillations

We illustrate that the collective electron oscillations induced by the laser pulses can be coherently controlled by varying the CEP of the laser pulses, directly in the time domain. A modulation of the CEP of the laser pulse controls the CEP of the laser-induced polarization, which in turn coherently modulates the CEP of the driven plasmon oscillations. As the technique presented in this work is a self-referencing technique, we probe linear plasmon oscillations induced in the junction of the nanoantennas by varying the CEP of one of the laser pulses (pulse-1 in Fig. 1d), while keeping the CEP of the other pulse fixed (pulse-2 in Fig. 1d). The CEP of the 1st order diffracted pulse (pulse-1) is controlled by varying the phase of the radio frequency phase-shifter (see section III in Supplementary Information) driving the AOFS. Fig. 5a shows the temporal evolutions of plasmon (electron) oscillations as a function of the varying CEP of pulse-1. Four representative traces at the CEP of 0, 0.5π, π, and 1.5π are shown in Fig. 5b. Phase control of plasmon (electron) oscillations as evident by a linear movement of the maxima of the oscillations on change of the CEP can be clearly seen in Fig. 5a, b.

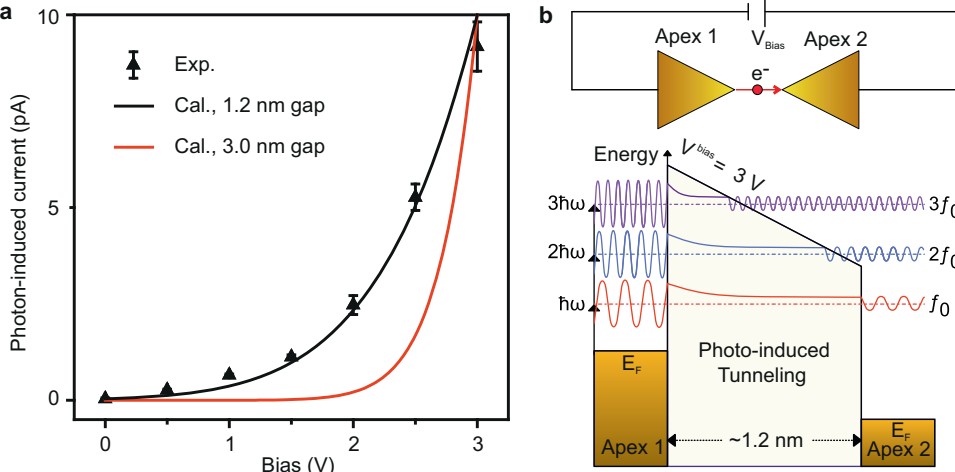

**Fig. 4 | Photo-assisted electron tunnelling in the nanoantenna junction. a** Black points: Variation of the photocurrent signal measured at the lock-in frequency of $f_0$ as a function of the increasing bias in the nanodevice. The pulse energy of the laser pulses was set at ~ 100 pJ and the delay between the pulses (pulse-1 and pulse-2) was set to zero. Black and red-curves show the calculated electron tunnelling probability, considering only single-photon excitation as a function of the increasing bias in the nanoantenna junction, where junction gaps of 1.2 and 3 nm are assumed, respectively. Error bars represent the standard deviation of the signal from ten consecutive measurements. **b** Bottom-panel: Schematic of the energy-level alignment in the biased nanodevice. Fermi level ($E_F$) of the Au nanoantenna on the left side (Apex 1, top-panel) is upshifted with respect to the Fermi level of the nanoantenna on the right side (Apex 2, top-panel). Electron oscillations above the Fermi level stimulated by one-, two- or three-photon absorption, can lead to photo-assisted electron tunnelling to the other side of the junction, as also schematically illustrated in the top-panel.

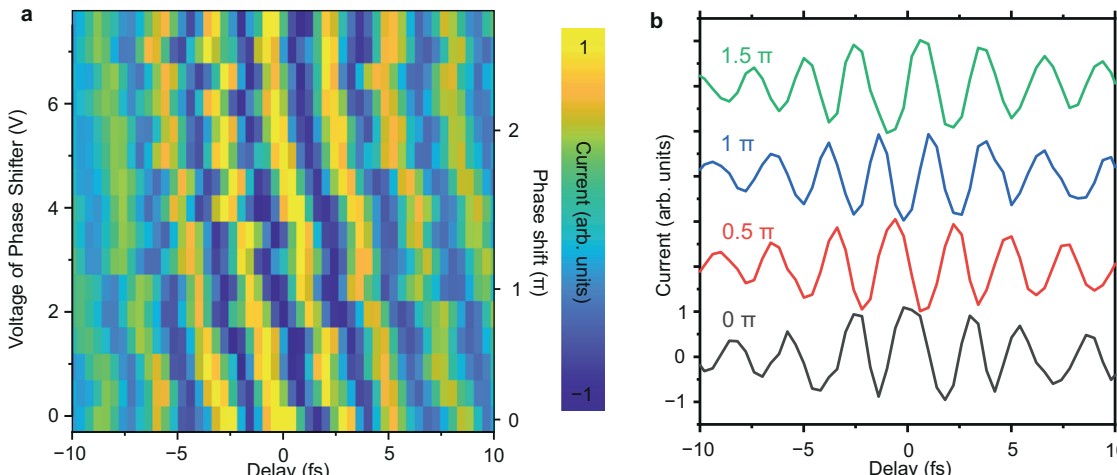

**Fig. 5 | Phase control of plasmon oscillations. a**, A series of time-resolved plasmon oscillations measured as a function of the CEP of pulse-1 in the nanodevice. The CEP of pulse-2 was kept fixed. Electron oscillations were sampled at the lock-in frequency of $f_0$. **b**, Measured temporal variation of electron oscillations from the measurement in **a** for four different CEPs of pulse-1. The CEPs of the pulse-1 are annotated on top of each curve. The traces are shifted vertically for clarity.

## Discussion

Direct measurement of light-waves can enable the study of quantum properties of ultrashort pulses, e.g. quantum fluctuations associated with the ground state of the electric field and squeezing of light. By utilizing the technique of optical homodyne beating, the electronic polarization response of a single molecule trapped in the junction of the nanodevice can be traced in the time-domain[47]; this polarization response will have the fingerprints of molecular electronic and vibrational levels. The capability to coherently trigger, measure and control collective electron oscillations and the associated coherent multiphoton processes opens new prospects for understanding strong light-matter interaction in solids directly in the time domain as well as pave the way towards on-chip light-wave electronics at petahertz switching and read-out frequency[31,48].

## Methods

### Nanodevice fabrication

The devices were fabricated on fused silica substrates using electron beam lithography, evaporation, and lift-off techniques. One device consists of ten arrays, each containing seven bowties with identically designed geometric dimensions and junctions, (see Fig. 1 in the main-text). In the design, the opposing isosceles triangles (triangle height of 300 nm, base of 250 nm) forming the bowtie are separated by gap sizes of 10 nm, 5 nm, 0 nm, and −5nm. The designed sizes of the junctions, e.g. 5 nm, are identical for all bowties in one array. Negative gap sizes indicate merged (overlapping) triangles. Please note that the fabricated gap sizes will differ from the designed values due to the proximity effect and exposure characteristics of the resist. Additionally, the fabricated junction gaps will differ in size within one array due

to the same reasons mentioned above. Thus, differently sized junction gaps and/or connected bowties may exist within the same array even though they have identical sizes in the design. The electrical connections are fanned out allowing to electrically connect and address every bowtie array individually. In addition, two bowtie arrays are replaced by a rectangle of 1 x 3.5 microns (short cut scenario) allowing for electrical reference measurements.

For nanofabrication, 80 nm of CSAR 62 resist (Allresist) was spin-coated on a fused silica substrate (10 mm x 10 mm) and baked for 60 s at 180° C. To avoid charging during electron exposure with the Raith Eline Plus system, a layer of ESPACER 300Z (Showa Denko, Singapore) was spin-coated (5000 rpm for 60 s) on top of the CSAR 62 resist. The patterning of the nanostructures (bowties and 30 nm narrow electrical connections, see electron micrograph in Fig. 1) was performed with an electron beam energy of 20 keV, a current of 0.02 nA and a dose of 130 $\mu C/cm^{-2}$. To fabricate nanometer-sized gaps, corrections for the proximity effect were included in the design process and the movement of the electron beam was optimized. The micro- and millimeter-sized structures (contact pads and electronic connections, see image in Fig. 1, main-text) were patterned in the same exposure step with the same energy and dose, but a larger electron current (9.5 nA). After exposure, the conducting ESPACER was removed in ultrapure water (2 s). Subsequently, the resist was developed in AR 600-546 (Allresist) for 60 s, stopped in AR 600-60 (Allresist) for 30 s and immersed in propan-2-ol for 30 s. Using electron beam evaporation at a pressure of $5 \times 10^{-7}$ mbar, a chromium (Cr) adhesion layer of 3 nm followed by 30 nm of gold (Au) has been deposited on the developed sample. Lift-off in N-Ethyl-2-pyrrolidon (Allresist) at 80 °C was performed for at least 8 h, to remove the gold-covered and non-exposed CSAR resist and reveal the fabricated structure.

### Experimental set-up

In our experiments, CEP-stable ultrashort laser pulses were split into two arms of the Mach-Zehnder interferometer (Fig. 1d in the main-text) by a beam splitter (BS). Laser pulses in one arm of the interferometer were loosely focused onto an acousto-optic-frequency-shifter (AOFS) by a bi-convex lens of a focal length of ~25 cm. A transverse radio frequency wave of frequency $f_r + f_o$ (~80 MHz + 700 Hz) runs through the AOFS with the average power of the incident laser pulses being ~300 mW. The fused silica AOFS of 20 mm thickness is driven by an arbitrary waveform synthesizer and the power of the radio frequency (RF) wave is amplified by an RF amplifier up to ~2 W. The temporal dispersion of the laser pulses acquired on traversing through the dispersive medium of the AOFS was pre-compensated by multiple reflections off the surface of two pairs of double-angled chirped dielectric mirrors before traversing through the AOFS. The $0^{th}$ order diffracted beam out of the AOFS is blocked, whereas the $1^{st}$ order frequency upshifted laser beam (see also section II) is combined with the laser pulses from the other arm of the interferometer. In order to have identical temporal profiles of the incident laser pulses from the two arms of the interferometer, a block of fused silica glass of ~20 mm thickness was added in the other beam path, (Fig. 1d, main-text) to induce the same amount of dispersion in the laser pulses as for the pulses traversing through the AOFS. The laser pulses were characterized with non-collinear second-harmonic frequency-resolved optical gating (FROG) measurements to make sure that the laser pulses incident on the bowtie nanoantennas were transform-limited in their temporal duration (see Fig. S1 in Supplementary Information).

The two combined laser pulses (pulse-1 and pulse-2 in Fig. 1d, main-text) are then focused by an off-axis parabolic mirror (OAPM) of focal length ~2.5 cm to the bowtie nanoantenna device mounted on a precision 3D stage. Markers in the nanodevice and a high zoom objective (10×) placed behind the nanodevice enable the precise positioning of the nanoantenna junction in the laser focal spot

(~10 μm diameter). The photo-assisted tunnelling current induced by the ultrashort laser pulses in the nanoantenna junction is amplified by a high gain (× $10^9 V/A$) current amplifier (Femto, DLPCA-200) and measured with a lock-in amplifier.

## Data availability
The data that support the findings of this study are available from the corresponding author on request.

## Code availability
The codes used for the DFT simulations in this study are available from the corresponding author on request.

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

## Acknowledgements

We thank Javier Aizpurua, Andrey Borissov and Shaoxiang Sheng for fruitful discussions, and Wolfgang Stiepany and Marko Memmler for technical support. A.M.J acknowledges the Alexander von Humboldt Foundation for financial support.

## Author contributions

M.G. conceived the project and designed the experiments. K.K. supervised the project. Y.L., M.G., A.M.J built the experimental set-up, performed the experiments and analyzed the experimental data. F.N. and N.L. contemplated the design of the nanodevices and fabricated the nanodevices. F.N. performed the finite element analysis simulations. All authors interpreted the results and contributed to the preparation of the manuscript.

## Funding

## Competing interests

The authors declare no competing interests.
