## [Peer Review File · Nature Communications]

Real-Time Tracking of Coherent Oscillations of Electrons in a Nanodevice by Photo-assisted TunnellingREVIEWER COMMENTS

Reviewer #1 (Remarks to the Author):

The paper by Garg and coworkers reports on a very elegant experimental method for characterizing the plasmonic response of a metal nanostructure, in particular a gold bowtie antenna. In this experiment an array of seven bowtie antennas is illuminated by two delayed ultrashort few-cycle laser pulses and the photo-assisted tunnelling current is recorded as a function of delay. In the so-called homodyne detection method one of the two pulses has carrier wave frequency shifted by a small amount f_0 using an acousto-optic phase-shifter, and the photocurrent demodulated at frequency f_0 and recorded as a function of the pulse delay allows one to determine amplitude and phase of the plasmonic response of the bowtie. By increasing the light intensity, also the nonlinear plasmonic response can be accessed, and finally it is also shown that the phase of the collective electronic oscillations can be controlled by changing the carrier-envelope phase (CEP) of one of the pulses.

This is a well-written paper that reports original and important results which deserve to be published in Nature Communications. I have however the following comments to the authors:

- i) I would put the experimental setup (currently only in the supporting information) in the main text, to better clarify the experiment; the description "The 0th and 1st order diffraction beams coming from the AOFS are slightly shifted in their carrier frequencies, by f_0 . These two laser pulses are then combined and focused in the junction of the nanoantenna..." is misleading. In fact it is not the 0th and 1st order which are focused, but the 1st order and a replica of the pulse obtained in a Mach-Zehnder interferometer;
- ii) figure 2c shows the comparison between the driving laser field and the plasmonic response, which is delayed by 3.2 fs; is such delay between the driving field and the response measurable also experimentally?
- iii) has the electric field of the pulse impinging on the bowtie antenna been determined, independently from the photocurrent measurement? It would be important to demonstrate that a perfectly transform-limited pulse is incident on the bowtie.
- iv) as a minor point, in figure S1 the Mach-Zehnder interferometer contains a block of glass in the arm that does not have the AOFS; its function is to compensate the dispersion introduced by the AOFS, but this should be explained a bit better and details on the block material and thickness should be given;
- v) finally it is shown that the phase of the tunnelling photocurrent can be changed by varying the CEP of one of the two pulses; I am not sure if this can be properly called "coherent control" in the sense given to this term in nonlinear spectroscopy, i.e. manipulating the spectral phase of a pulse or a pulse sequence in order to control the efficiency of a photoinduced process.

Reviewer #2 (Remarks to the Author):

The authors introduce a homodyne detection technique to measure linear and nonlinear photocurrents in nanoantennas. They measure photocurrents from a bowtie nanoantenna driven by two phase locked light fields. One of the fields is frequency shifted using an acousto-optic modulator.

They use the method to measure the linear response of the antenna in the time domain and show that the local field can be phase-shifted by controlling the carrier envelope phase of the laser. They also begin to see higher order nonlinearities in the photocurrent, and demonstrate the onset of nonlinear photocurrent generation. This is a new and interesting finding.

The paper is well and clearly written and in my opinion well suited for Nature Communications. I essentially have three questions:

- a) The authors state they start to observe signatures of photodegradation at pulse energies of around 200 pJ. How do these damage thresholds compare to far-field experiments where nonlinear

photoemission in the weak and strong field regime has now been studied by a number of research groups. Is the damage threshold severely affected by the presence of the STM tip?

b) What is the local field amplitude in the gap of the STM tip. How does the local field amplitude compare to that in far field strong field emission experiments where the onset of multiphoton and strong field emission is typically seen at local field amplitudes in the range of 5 – 15 V/nm.

c) The authors use a phenomenological model to interpret their experiments. They introduce an „effective Fermi electron distribution“ for this. Their 7-fs laser will impulsively excite the electron gas and create (most likely) a non-equilibrium distribution that does, initially, not match a Fermi distribution (see e.g., Ropers et al., PRL (2007) or experiments by Fabrice Vall’ee). Hence it is not so obvious that a simple Fermi distribution can reproduce the experiments. What are the assumptions of the model? What is the temperature of the electron gas in the introduced Fermi model?

Reviewer #3 (Remarks to the Author):

The entire flow of the arguments in the paper is based on the idea of the laser-assisted tunneling, where optically excited electrons traverse the gap in the bowtie nanoantenna thanks to the barrier reduction by an applied bias.

Any further discussion of this work is premature before the authors present a clear measurement of the gap size. Indeed, in the beginning of the paper the authors discuss the gap of "only a few nm". In the Supplementary material they state: "The bowtie are separated by the gap sizes of 10 nm, 5 nm, 0 nm, and -5 nm. The designed sizes of the junctions, e.g. 5 nm, are identical (!!!) for all bowties in the array." This gives an impression of the extreme control of the geometry, which is apparently far from the reality. Indeed, at the end of the paper, in order to explain the experimental data, the gap size of 1.2 nm was used. Some hand-waving discussion is then presented to say that actually the gap size is pretty much unknown. But this is an extremely important parameter of the theory validating or invalidating the explanation. Thus, using the gap of 5 nm would result in the step function in the photoinduced current dependence on the bias. Obviously, this would not correspond to an experiment. This is not a surprise because 50 Angstrom gap is way too large to talk about tunneling. In this situation one would rather think of multi-photon emission followed by ballistic transport, but given the work function of Gold this would require clearly more than 2 photons.

By the way, can the authors show the results measured obtained with different gap sizes of -5, 0, +5 and +10 nm. Or at least measured with device where +10 nm was targeted for nano-fabrication?. This would be a clear test of the proposed theory.

In any case: the gap size has to be clearly determined before any further consideration of the paper.

Additional comments.

1. What is the contribution of different multi-photon processes in Fig. 4 and how it corresponds to the experimental data in Figure 3e?

2. How the occupations were determined in Eq. 17? Without clear explanation of the corresponding calculations, it is just a fit parameter and the agreement with experiment can not be considered as a proof of the model validity.

3. The authors correlate the nonlinear dependence of the current through the junction with nonlinear polarization density. What do they mean by that considering the vacuum gap? Why one can not simply state that as any nonlinear signal the current dependence on the driving field

should be given by the Taylor series? I find the speculation on the polarization misleading.

4. The time delay of 10 fs in pumping the energy from the laser pulse to plasmon oscillations corresponds to the width of the plasmon resonance of approx 0.065 eV. This is not the calculated width of the plasmon resonance (see figure 2b). In figure 2c (simulations) I can read more 3.2 fs, actually it is much closer to the measured and calculated width in figure 2b. Why 10 fs delay is quoted in the Abstract?

Reviewer #4 (Remarks to the Author):

The authors demonstrate results of petahertz-domain frequency sampling using photo-induced current across a gold nanoantennae gapped device. They show that photocurrents in this device resulting from linear and higher order contributions to the electron oscillations can be disentangled using a homodyne detection technique to resolve CEP of the incident light wave. Unfortunately, this result is at best an incremental development on significant previous work in this field, and therefore I do not recommend it for publication in Nature Communications.

The introduction, especially paragraph 2, claims this is a novel approach, but then the authors cite papers that perform this exact technique to measure the same phenomena (Blochl source 11, Yang source 47, Ritzkowsky et. al uncited, Zimin et. al uncited). In fact, the device setup and resulting measured photocurrent matches exactly to the work of Yang and coworkers, with the only difference being the use of an AOFS instead of wedges for phase delay. The detection and correction for higher order frequency components is demonstrated by Zimin and coworkers, who employ a similar Mach-Zender configuration for the incident light pulses. To claim novelty is misleading and these claims must be removed from the manuscript.

Further, these previous works include comparisons between different device configurations, whereas the current manuscript only includes one device. The authors must think critically about how to use these established techniques to demonstrate novel results. For example, the inclusion of a trapped single molecule to determine its vibrational spectrum, as suggested in the conclusion, would be a significant result.

Nonetheless, the organization and descriptions found in the manuscript and figures are presented clearly and the experimental results and modelling are high quality. Changes to the introduction, motivation, and a more accurate citation of previous results and how this work fits into the field are a necessity. Even then, with the current results the manuscript would not be on the level of a Nature Communications article and should be submitted to a more specialized journal.

Minor: Figure 2c caption says "black double arrow" when the arrow is gray.

References:

Ritzkowsky, F. et al. Tailoring the impulse response of petahertz optical field-sampling devices. Int. Conf. Ultrafast Phenom. (UP), Optica Tech. Digest, Th1A.4 (2022).
Zimin, D. et al. Petahertz-scale nonlinear photoconductive sampling in air. Optica 8(5), 586-590 (2021).

Response to the referees

We would like to cordially thank all the referees for their invaluable comments. We have revised the manuscript in accordance with their suggestions and criticisms. We mark essential revisions as **REV#** and have underlined the new text (____) in the main text and supplementary materials (SM) to ease the evaluation.

We hope that the referees will acknowledge the significant improvements made in the manuscript, certainly driven by their keen comments, and will now be happy to recommend it for publication in Nature Communications. Below we enlist detailed responses to all comments of the referees.

Reviewer #1 (Remarks to the Author):

The paper by Garg and coworkers reports on a very elegant experimental method for characterizing the plasmonic response of a metal nanostructure, in particular a gold bowtie antenna. In this experiment an array of seven bowtie antennas is illuminated by two delayed ultrashort few-cycle laser pulses and the photo-assisted tunnelling current is recorded as a function of delay. In the so-called homodyne detection method one of the two pulses has carrier wave frequency shifted by a small amount f_0 using an acousto-optic phase-shifter, and the photocurrent demodulated at frequency f_0 and recorded as a function of the pulse delay allows one to determine amplitude and phase of the plasmonic response of the bowtie. By increasing the light intensity, also the nonlinear plasmonic response can be accessed, and finally it is also shown that the phase of the collective electronic oscillations can be controlled by changing the carrier-envelope phase (CEP) of one of the pulses.

This is a well-written paper that reports original and important results which deserve to be published in Nature Communications. I have however the following comments to the authors:

We would like to sincerely thank the reviewer for his/her kind words and for recognizing a series of novel elements in our work. We are glad to make all possible efforts to increase the clarity of all the points raised and to meet the expectations of the reviewer.

i) I would put the experimental setup (currently only in the supporting information) in the main text, to better clarify the experiment; the description "The 0th and 1st order diffraction beams coming from the AOFS are slightly shifted in their carrier frequencies, by f_0 . These two laser pulses are then combined and focused in the junction of the nanoantenna..." is misleading. In fact it is not the 0th and 1st order which are focused, but the 1st order and a replica of the pulse obtained in a Mach-Zehnder interferometer;

We would like to thank the referee for this suggestion. In the revised manuscript, we have included the experimental setup in the main-text, please see Fig.1d, and have removed the former Fig. S1 from the SM. We have also adjusted the descriptions accordingly. Please see **REV#1** in the main-text.

Revised Fig.1. Complete experimental set-up is now shown in the panel d.

ii) figure 2c shows the comparison between the driving laser field and the plasmonic response, which is delayed by 3.2 fs; is such delay between the driving field and the response measurable also experimentally?

The delayed response of the plasmon oscillations as shown in Fig. 2c is captured by our simulations. In the experiments, we have used the homodyne beating technique to measure the local plasmonic response in the nanoantenna junction. Although the spectrum and the phase of the plasmon oscillations were captured, the relative delay of the plasmon oscillations with respect to the incident electric field cannot be measured.

iii) has the electric field of the pulse impinging on the bowtie antenna been determined, independently from the photocurrent measurement? It would be important to demonstrate that a perfectly transform-limited pulse is incident on the bowtie.

The laser pulses were characterized with second-harmonic (non-collinear) frequency-resolved optical gating (FROG) measurements. The experimental and reconstructed FROG traces of the

original and the 1st order laser pulses are now included in the SM, please see **REV#2** in the SM. The FROG measurements show that the incident laser pulses from both arms of our Mach-Zehnder interferometer on the bowtie nanoantennas were transform-limited. Please note that the transform-limited pulse duration of ~ 4.5 fs is the FWHM of the temporal intensity profile of the pulses. In terms of the electric field, the pulse durations will be $\sqrt{2}$ times higher, which would make the pulse duration to be ~ 7 fs, as mentioned in the main-text.

Fig. R1 | FROG measurements and reconstructions for pulse-2 (a, the original laser beam) and pulse-1 (1st order diffraction beam, b). First Column: The experimental FROG traces. Second Column: The reconstructed FROG traces. Third Column: The temporal profiles of laser intensity (red-curve) and phase (blue-curve). Fourth Column: The spectral profiles of laser intensity (blue-curve) and phase (red-curve). The reduced bandwidth of the laser pulses as retrieved from the FROG measurements is due to the limited spectral phase matching bandwidth of the ~ 10 μm thick BBO crystal as used in the FROG set-up. Please see **REV#2** in SM.

iv) as a minor point, in figure S1 the Mach-Zehnder interferometer contains a block of glass in the arm that does not have the AOFS; its function is to compensate the dispersion introduced by the AOFS, but this should be explained a bit better and details on the block material and thickness should be given;

We thank the referee for making this suggestion. A more detailed description of the experimental setup is now included in the SM. Please see **REV#3** in SM.

“The temporal dispersion of the laser pulses acquired on traversing through the dispersive medium of the AOFS was pre-compensated by multiple reflections off the surface of two pairs of double-angled chirped dielectric mirrors before traversing through the AOFS In order to

have identical temporal profiles of the incident laser pulses from the two arms of the interferometer, a block of fused silica glass of ~ 20 mm thickness was added in the other beam path, (Fig. 1d, main-text) to induce the same amount of dispersion in the laser pulses as for the pulses traversing through the AOFS.”

v) finally it is shown that the phase of the tunnelling photocurrent can be changed by varying the CEP of one of the two pulses; I am not sure if this can be properly called "coherent control" in the sense given to this term in nonlinear spectroscopy, i.e. manipulating the spectral phase of a pulse or a pulse sequence in order to control the efficiency of a photoinduced process.

We agree with the referee that in "*coherent control*" the efficiency (yield) of the photo-induced processes should be controlled by tuning the phase of the pulses. In the current work, we demonstrated that the phase of the local plasmon oscillations in the nanoantenna junction can be controlled by varying the phase of the incident laser pulses (Fig. 1d). **We have now changed the term “*coherent control*” to “*phase control*” in the revised manuscript.** Please see **REV#4** in the main-text.

We sincerely thank the referee for pointing out some weaknesses in our presentation. The suggestions by the referee have certainly allowed us to strengthen the presentation and the conclusions of our work.

Reviewer #2 (Remarks to the Author):

The authors introduce a homodyne detection technique to measure linear and nonlinear photocurrents in nanoantennas. They measure photocurrents from a bowtie nanoantenna driven by two phase locked light fields. One of the fields is frequency shifted using an acousto-optic modulator.

They use the method to measure the linear response of the antenna in the time domain and show that the local field can be phase-shifted by controlling the carrier envelope phase of the laser. They also begin to see higher order nonlinearities in the photocurrent, and demonstrate the onset of nonlinear photocurrent generation. This is a new and interesting finding.

The paper is well and clearly written and in my opinion well suited for Nature Communications. I essentially have three questions:

We would like to sincerely thank the reviewer for his/her kind words and for recognizing a series of novel elements in our work. We are glad to make all possible efforts to increase the clarity of all the points raised and to meet the expectations of the reviewer.

a) The authors state they start to observe signatures of photodegradation at pulse energies of around 200 pJ. How do these damage thresholds compare to far-field experiments where nonlinear photoemission in the weak and strong field regime has now been studied by a number of research groups. Is the damage threshold severely affected by the presence of the STM tip?

In the near-field photoemission experiments conducted on nanoparticles or free-standing nanotips, the maximum incident laser intensity is typically maintained below $10^{12} \text{ W cm}^{-2}$. This is related to the damage threshold of materials, as outlined in earlier studies (*Phys. Rev. Lett.* **105**, 147601 (2010); *Rev. Mod. Phys.* **92**, 025003, (2020)).

In contrast to earlier near-field experiments, the present study reveals a notably lower damage threshold, specifically at incident laser intensities of below $\sim 10^9 \text{ W cm}^{-2}$. Comparable damage thresholds attributed to processes such as electromigration or field evaporation have been observed in other **on-chip** experiments where the near-field photoemission processes are detected by measuring a photocurrent (*Nat. Phys.* **13**, 335, (2017); *Nat. Commun.* **11**, 3407, (2020)). Systematic studies of the on-chip nanoantenna device degradation during photoemission measurements have been performed in *Nat. Commun.* **11**, 3407 (2020). **In our work, since we are measuring in the photon-driven regime, electromigration induced by the extremely high transient current densities could be the primary reason for a significantly lower threshold of photo degradation.**

Ultrahigh-vacuum low-temperature STM provides stable conditions to investigate the nonlinear photoemission phenomenon (*New J. Phys.* **22**, 033047, 2020). Optical laser pulses lasting only a few fs have been integrated with an STM to study the field-driven processes (*Science* **367**, 411, (2020); *ACS Nano* **16**, 1447, (2022)). Incident laser pulse intensities in the range of $\sim 10^{11} \text{ W cm}^{-2}$ were used in these works, which is close to those used in other near-field (free-standing nanotip)

experiments. **In our current experiments, conducted in ambient conditions, the nanoantenna bowtie structures are less stable upon illumination with strong laser pulses.**

In the revised manuscript, we have added one sentence discussing the possible reasons for the photodegradation:

“However, the measurements at such high laser pulse energies are not very stable as the nanodevice is prone to physical damage, which can be induced by electromigration or field evaporation [Ref. 42].” Please see **REV#5** in the main-text.

b) What is the local field amplitude in the gap of the STM tip. How does the local field amplitude compare to that in far field strong field emission experiments where the onset of multiphoton and strong field emission is typically seen at local field amplitudes in the range of 5 – 15 V/nm.

We kindly note that the tunnelling currents in the current work were measured in a bowtie nanoantenna junction, which is similar to a plasmonic nanocavity of an STM junction.

The maximum free-space peak electric field strength used in our current work is estimated to be ~ 0.12 V/nm, which is much smaller compared to the two other works demonstrating field-driven photoemission from on-chip resonant nanoantenna devices: 0.64 V/nm in Ref. 8 (*Nat. Photon.* **15**, 787-787 (2021)), and up to 1 V/nm in Ref. 15 (*Nat. Phys.* **16**, 341-345 (2020)).

Although the free-space peak electric field strength can be easily estimated, the exact local field amplitude in the nanoantenna junction is hard to determine since the enhancement factor of the localized surface plasmons is experimentally unknown.

In a work by another group, by measuring the laser-induced photocurrent in a bowtie nanodevice (*Nature Physics* **16**, 341, (2020)), an 11-fold enhancement of the free-space electric field in the junction of the nano-device was estimated. In another work of on surface-enhanced Raman spectroscopy using bowtie structures (*J. Opt.* **17**, 125002 (2015)), the experimentally measured enhancement factor is around 6-10, depending on the local geometry of the nanostructure. Thus, we estimate the peak local field amplitude in the nanoantenna junction to be below 1.2 V/nm, considering a field enhancement factor of 10. This peak field strength is much smaller compared to the field strengths in the far-field strong-field emission experiments, as well as the near-field (free-standing tips) strong-field emission experiments.

c) The authors use a phenomenological model to interpret their experiments. They introduce an „effective Fermi electron distribution“ for this. Their 7-fs laser will impulsively excite the electron gas and create (most likely) a non-equilibrium distribution that does, initially, not match a Fermi distribution (see e.g., Ropers et al., PRL (2007) or experiments by Fabrice Vall’ee). Hence it is not so obvious that a simple Fermi distribution can reproduce the experiments. What are the assumptions of the model? What is the temperature of the electron gas in the introduced Fermi model?

We totally agree with the reviewer that the 7-fs laser pulses will create a non-equilibrium distribution of photoelectrons from the Fermi energy level up to $E_j = j\hbar\omega$ above the Fermi level (where $j = 1$ or 2 , corresponding to the number of photons absorbed by the electrons). Following Ref. 46 (*New J. Phys.* **22**, 033047 (2020)), we have included the energy widths δE_j in equation 17 of the SM to account for the possible energy redistribution by the effects of thermalization. Since the energy broadening is much smaller compared with the photo energy (~ 1.6 eV), we have assumed $\delta E_j \sim 0$ (zero temperature). That is, we have assumed $f_{eff}(E) = \Delta_j$ when $E \leq E_j$, and $f_{eff}(E) = 0$ when $E > E_j$. In the revised SM, we have included a detailed description of the tunnelling model. Please see **REV#6** in SM.

We sincerely thank the referee for his/her careful review of this work. The suggestions by the referee have certainly allowed us to strengthen the presentation and the conclusions of our work.

Reviewer #3 (Remarks to the Author):

The entire flow of the arguments in the paper is based on the idea of the laser-assisted tunneling, where optically excited electrons traverse the gap in the bowtie nanoantenna thanks to the barrier reduction by an applied bias.

Any further discussion of this work is premature before the authors present a clear measurement of the gap size. Indeed, in the beginning of the paper the authors discuss the gap of "only a few nm". In the Supplementary material they state: "The bowtie are separated by the gap sizes of 10 nm, 5 nm, 0 nm, and -5 nm. The designed sizes of the junctions, e.g. 5 nm, are identical (!!!) for all bowties in the array." This gives an impression of the extreme control of the geometry, which is apparently far from the reality. Indeed, at the end of the paper, in order to explain the experimental data, the gap size of 1.2 nm was used. Some hand-waving discussion is then presented to say that actually the gap size is pretty much unknown. But this is an extremely important parameter of the theory validating or invalidating the explanation. Thus, using the gap of 5 nm would result in the step function in the photoinduced current dependence on the bias. Obviously, this would not correspond to an experiment. This is not a surprise because 50 Angstrom gap is way too large to talk about tunneling. In this situation one would rather think of multi-photon emission followed by ballistic transport, but given the work function of Gold this would require clearly more than 2 photons.

We completely agree with the reviewer that the gap size is an extremely important parameter in our studies. To clarify this issue, we **(i) rephrased the sections** in the manuscript and supporting information on the device fabrication and gap size determination with SEM and **(ii) performed new experiments using single bowtie nanoantennas.**

(i) As mentioned by the reviewer, the fabrication of seven bowties featuring identical sub 10 nm sized gaps within one array is not possible. Such a reproducible control of the junction sizes cannot be achieved with our EBL-based fabrication method. We tried to convey this by distinguishing "**designed**" and "**fabricated**" junctions already in the supporting information of the submitted manuscript. To clarify the valid concern and avoid misunderstandings, we added the following sentences to the revised supplement: "*Additionally, the fabricated junctions will differ in size within one array due to the same reasons mentioned above. Thus, differently sized junction gaps and/or connected bowties may exist within the same array even though they have the identical sizes in the design.*" Please see **REV#7** in section 1.1 of the SM.

We also removed the following misleading sentence: "*The designed sizes of the junctions, e.g. 5 nm, are identical for all bowties in one array.*"

In the main-text we rephrased the seemingly hand-waving discussion and pointed out that the gap sizes cannot be determined with the available scanning electron microscope. "*Using scanning electron microscopy, we found that the fabricated junction gap of a single bowtie can be as small as 8 nm. Smaller gap sizes may occur but could not be resolved with the available SEM. However, the actual junction gap can be smaller due to electromigration of Au atoms in the nanoantenna junction in the device.*" Please see **REV#8** in the main-text.

(ii) We fabricated and measured single bowties (Fig. R2) instead of arrays composed of seven bowties as presented in the previous submission. Compared to the seven bowtie arrays, **single bowties have the advantage of featuring only one gap size**. The size of the fabricated junction still cannot be precisely controlled and may differ from the designed values due to the proximity effects and exposure characteristics of the resist. However, signals and effects arising from differently-sized junction gaps can be excluded. Also, for single antennas, EBL-based nanofabrication as described in the supporting information (section 1.1), was applied.

Asymmetry in the coherent electron oscillations similar to the 7-bowtie structure (Fig. 2a, main-text) was also measured for the single bowtie nanoantenna structure. **A phase shift of π radians across the spectral bandwidth** of the local plasmonic response was also measured for the single bowtie structure (dashed blue-curve in Fig. R2b, right-panel).

We have now included these new measurements with the single bowtie nanoantennas in the SM. Please **REV#9** in the main-text and SM (Fig. S5).

Fig. R2 | Real-time tracking of coherent collective electron oscillations in a single bowtie nanoantenna. **a**, Representative scanning electron microscope (SEM) images of a single bowtie nanoantenna with a designed gap size of -5 nm. **b**, Left-panel: Variation of the laser-induced photocurrent as a function of the delay between pulse-1 and pulse-2 laser pulses of slightly different carrier frequencies. The applied DC bias in the nanoantenna junction is 3 V, and the

incident laser pulse energy is 100 pJ. Right-panel: The spectrum and phase of experimentally measured local plasmon oscillations in the nanoantenna junction. The dotted black-curve shows the spectrum of the incident laser pulses on the nanoantenna junction. **c**, Variation of the photocurrent in the nanoantenna junction as a function of the applied DC bias. The red curve shows the calculated electron tunnelling probability considering only single-photon excitation with a junction gap (tunnelling gap) of 0.7 nm.

Regarding the interpretation of the photocurrent generation in the nanodevices, we can eliminate two alternative mechanisms – (1) multi-photon emission and (2) optical field emission. As the referee correctly mentioned, multi-photon emission requires three- or four-photon absorption, which would show a much more nonlinear behavior in the laser pulse energy scaling experiments as shown in Fig. 3 of the main-text. The photocurrent supported by ballistic transport would be linear with respect to the applied bias voltages. This is contrary to the experimental observations of quadratic power-dependent behavior and nonlinear bias-dependent behavior. The other mechanism of optical field emission can be achieved at a much higher peak incident electric field strengths, of 0.64 V/nm in Ref. 8 (*Nat. Photon.* **15**, 787-787 (2021)), and up to 1 V/nm in Ref. 15 (*Nat. Phys.* **16**, 341-345 (2020)). In the current work, we are using weak electric fields only up to ~ 0.12 V/nm.

The only plausible explanation for the photocurrent generation in our experiments is the mechanism of photo-assisted electron tunnelling. **This mechanism aligns with the understanding of photo-assisted tunnelling current in the scanning tunnelling microscope-based experiments, with a tunnelling gap size of ~ 1 nm** (Ref. 46, *New J. Phys.* **22**, 033047 (2020)).

By the way, can the authors show the results measured obtained with different gap sizes of -5, 0, +5 and +10 nm. Or at least measured with device where +10 nm was targeted for nano-fabrication?. This would be a clear test of the proposed theory.

In Fig. R4, we show time traces of the laser-induced photocurrents from two different nanodevices with the same designed junction gap sizes of 10 nm. The measured photocurrent as a function of the delay between two laser pulses shows clear oscillations for the first nanodevice (A-1), whereas the oscillations for the second nanodevice (A-2) are not clear. This difference is attributed to the difficulty of achieving precise control over the fabricated nanoantenna junction gap and the local atomic scale structure at the apexes of the nanoantennas.

Measuring I-V curve (Current Vs. Voltage) is the only reliable way to determine junction gaps (< 2 nm) which cannot be measured by state-of-the-art SEM techniques. Our SEM has a spatial resolution of ~ 8 nm. We have used the most reliable approach of I-V curve measurement to precisely determine the local nanoantenna junction gaps.

Fig. R4 | Variation of the laser-induced photocurrent as a function of the delay between pulse-1 and pulse-2 laser pulses measured for two nanodevices with the same designed junction gap size of 10 nm. The time trace in **a** is measured at the bias voltage of 3 V, and the incident laser power of 150 pJ. The time trace in **b** is measured at 6 V and 200 pJ.

In any case: the gap size has to be clearly determined before any further consideration of the paper.

Additional comments.

1. What is the contribution of different multi-photon processes in Fig. 4 and how it corresponds to the experimental data in Figure 3e?

The bias dependence measurement shown in Fig. 4 (main-text) was conducted at the incident laser pulse energy of ~ 100 pJ, where the contribution of 2-photon excitations is low (around noise level). Please see the laser pulse energy scaling experiment shown in Fig. 3e.

2. How the occupations were determined in Eq. 17? Without clear explanation of the corresponding calculations, it is just a fit parameter and the agreement with experiment can not be considered as a proof of the model validity.

The ultrafast laser pulses will create a non-equilibrium distribution of photoelectrons from the Fermi energy level up to $E_j = j\hbar\omega$ above the Fermi level (where j is the number of photons absorbed by the electrons). Following Ref. 46 (*New J. Phys.* **22**, 033047 (2020)), we have induced the energy widths δE_j in equation 17 of the SM to account for the possible energy redistribution by thermalization. Since the energy broadening is much smaller compared with the photo energy (~ 1.6 eV), we have assumed $\delta E_j \sim 0$ (zero temperature). That is, we have assumed $f_{\text{eff}}(E) = \Delta_j$ when $E \leq E_j$, and $f_{\text{eff}}(E) = 0$ when $E > E_j$.

We agree with the referee that a fitting curve can't guarantee the validity of the model. **The underlying mechanism of photo-assisted electron tunnelling is proposed by eliminating**

the other possible mechanisms such as multi-photon emission (followed by ballistic transport) and optical field emission. The tunnelling model has provided additional support to our understanding. In the revised SM, a more detailed discussion of the tunnelling model is provided. Please see **REV#6** in the SM.

3. The authors correlate the nonlinear dependence of the current through the junction with nonlinear polarization density. What do they mean by that considering the vacuum gap? Why one can not simply state that as any nonlinear signal the current dependence on the driving field should be given by the Taylor series? I find the speculation on the polarization misleading.

The term “polarization” response (linear as well as non-linear) has been frequently used to describe the response of a material exposed to optical fields, as used in Ref. 6 (*Rev. Mod. Phys.* 92 (2020)). Indeed, it can be expressed as a Taylor series of the electric fields in the weak-field regime (perturbative regime). In the current work, the nonlinear current is explained with the nonlinear polarization response, to emphasize the coherent nature of multi-photon-assisted electron tunnelling processes in the nanoantenna junction. This description is still within the perturbative regime of interaction. A similar concept of **“coherent polarization”** was adopted to explain the multiphoton photoemission from metal surfaces in Ref. 41 (*Phys. Rev. X* 9, 011044 (2019)).

The term **“the vacuum tunnelling gap”** was used incorrectly, and now it’s changed to **“the tunnelling gap”** in the revised SM.

4. The time delay of 10 fs in pumping the energy from the laser pulse to plasmon oscillations corresponds to the width of the plasmon resonance of approx 0.065 eV. This is not the calculated width of the plasmon resonance (see figure 2b). In figure 2c (simulations) I can read more 3.2 fs, actually it is much closer to the measured and calculated width in figure 2b. Why 10 fs delay is quoted in the Abstract?

In a nanostructure, the response function, R , of the localized surface plasmons as a function of the frequency ω can be expressed as: $R(\omega) = \frac{1}{\omega_0^2 + 2i\gamma\omega - \omega^2}$, where ω_0 is the resonance angular frequency, and γ is the damping rate. In our simulation, we considered $\gamma = 0.12 \text{ fs}^{-1}$, that is, a time constant of $\sim 8.3 \text{ fs}$ or an energy of $\sim 80 \text{ meV}$. The simulated intensity spectrum of localized surface plasmons is expressed as $|R|^2$, which shows a resonant peak at ω_0 with a **bandwidth of 2γ** . Therefore, we estimate a bandwidth of $\sim 160 \text{ meV}$, which is consistent with our experimental results and the calculations using finite element simulations. A detailed description of the model can be found in ref. 40 (*Opt. Mat. Exp.* 11, 2817-2827 (2021)).

This estimated time constant of $\sim 8 \text{ fs}$ is indeed the T_2 decay time (*Phys. Rev. Lett.* 84, 5644, (2000)), which is also referred to as dephasing time in literature. The T_2 time can be extracted from the homogeneous linewidth Γ of the plasmonic response: $T_2 = 2\hbar/\Gamma$. The homogeneous linewidth of 160 meV corresponds to a T_2 time of 8 fs. To avoid any ambiguity, we have changed **“decay time”** to **“ T_2 decay time”** in the revised manuscripts. We have also added a sentence

estimating the spectral linewidth from the damping rate, and changed the “10 fs” decay time to “8 fs”, which comes directly from the damping rate used in the simulations. Please see **REV#10** in the main-text.

We are hopeful that the reviewer will now be convinced by our detailed response to his/her comments and will be happy to recommend our work for publication in Nature Communications.

Reviewer #4 (Remarks to the Author):

The authors demonstrate results of petahertz-domain frequency sampling using photo-induced current across a gold nanoantennae gapped device. They show that photocurrents in this device resulting from linear and higher order contributions to the electron oscillations can be disentangled using a homodyne detection technique to resolve CEP of the incident light wave. Unfortunately, this result is at best an incremental development on significant previous work in this field, and therefore I do not recommend it for publication in Nature Communications.

We would like to thank the reviewer for his/her time to work on our manuscript. The current work introduces and uses the technique of optical homodyne beating to sample the collective electron oscillations induced by ultrashort laser pulses in a nanostructure (localized surface plasmons). Our experiments capture the amplitude and phase of the plasmon oscillations, which can be explained by a simple damped harmonic oscillator model. Moreover, we also demonstrate that nonlinear electron oscillations induced in the nanostructure at higher incident intensities of the laser pulse can be tracked by our technique in real-time. Finally, we demonstrated a phase control of the local plasmon oscillations, directly in the time domain. **These findings in our work have not been reported by any other group. The novelty of the work is transparently recognized in the reports by reviewers #1 and #2.**

In the introductory sections, our work has been put in perspective on tracing local electron oscillations in a nanostructure (directly in the time domain). Earlier studies which incorporated the on-chip device design that we have also employed, have been properly cited. However, using comparable sample designs as in earlier works does not bring down the novelty of the current work. **The novelty of our contribution is defined by the experimental observations and the understanding of the time-resolved photocurrents, rather than the specific sample or device used in the work.**

The introduction, especially paragraph 2, claims this is a novel approach, but then the authors cite papers that perform this exact technique to measure the same phenomena (Blochl source 11, Yang source 47, Ritzkowsky et. al uncited, Zimin et. al uncited). In fact, the device setup and resulting measured photocurrent matches exactly to the work of Yang and coworkers, with the only difference being the use of an AOFS instead of wedges for phase delay. The detection and correction for higher order frequency components is demonstrated by Zimin and coworkers, who employ a similar Mach-Zender configuration for the incident light pulses. To claim novelty is misleading and these claims must be removed from the manuscript.

In our work, we have cited all the necessary references which have an overlap, e.g., in the device design or at least help us compare our work with the current state-of-the-art. A review of what other groups have done so far is definitely not the purpose of our work. We have already explained in the response to the previous point what the achievements of our work are. **A similar device design in any work (not only in the current work) definitely does not lower its novelty.**

The reviewer correctly mentions that we have used an AOFS in our work, we kindly ask the reviewer to acknowledge the advance this new development is able to bring forth. Of course, the

phase of the laser pulses can be modulated with the help of the wedges as with the help of an AOFS. **However, optical homodyne beating realized with an AOFS can do much more and really track coherent electron oscillations (linear as well as nonlinear) in a nanodevice.**

The work of Zimin et al. is a photoconductive sampling technique, where strong injection pulses generate photocarriers in air/quartz, which are then driven by a weaker and orthogonally polarized replica of the same pulse. This technique is commonly used for sampling weak THz pulses (Auston switches), where the photocarriers in the photoconductive switch/antenna are generated by an optical pulse (Auston et al. Appl. Phys. Lett. **37**, 371–373 (1980)). Zimin et al., have pushed the limit of photoconductive sampling to the NIR spectral range from the THz spectral range. Nevertheless, **we find it hard to contemplate the similarity between the work of Zimin et al. and our current work.**

Further, these previous works include comparisons between different device configurations, whereas the current manuscript only includes one device. The authors must think critically about how to use these established techniques to demonstrate novel results. For example, the inclusion of a trapped single molecule to determine its vibrational spectrum, as suggested in the conclusion, would be a significant result.

In the revised manuscript, we have performed experiments with a new device configuration: **single bowtie nanoantenna devices** (Fig. R2). **Asymmetry in the coherent electron oscillations** similar to the 7-bowtie structure (Fig. 2a, main-text) was also measured for the single bowtie nanoantenna structure. **A phase shift of π radians across the spectral bandwidth** of the local plasmonic response was also measured for the single bow-tie structure (dashed red-curve in Fig. R2).

We have now included these new measurements with the single bowtie nanoantennas in the SM. Please **REV#9** in the main-text and SM (**Fig. S5**).

Fig. R2 | Real-time tracking of coherent collective electron oscillations in a single bowtie nanoantenna. **a**, Scanning electron microscope (SEM) images of single bowtie nanoantenna. **b**, Left-panel: Variation of the laser-induced photocurrent as a function of the delay between pulse-1 and pulse-2 laser pulses of slightly different carrier frequencies. The applied DC bias in the nanoantenna junction is 3 V, and the incident laser pulse energy is 100 pJ. Right-panel: The spectrum and phase of experimentally measured local plasmon oscillations in the nanoantenna junction. The dotted black-curve shows the spectrum of the incident laser pulses on the nanoantenna junction. **c**, Variation of the photocurrent in the nanoantenna junction as a function of the applied DC bias. The red curve shows the calculated electron tunnelling probability considering only single-photon excitation with a junction gap (tunnelling gap) of 0.7 nm.

Tracking the polarization response of a single molecule trapped in the nanoantenna junction is definitely the next step of the present work. Undoubtedly, accomplishing this would indeed be a major advance in single-molecule studies, which nonetheless, depends on the significant and novel advances presented in the current work. Single-molecule studies are currently ongoing and including any of the results from these experiments would defocus from the key claims, techniques, and results presented in the current work. We hope for the understanding of the reviewer.

Nonetheless, the organization and descriptions found in the manuscript and figures are presented clearly and the experimental results and modelling are high quality. Changes to the introduction, motivation, and a more accurate citation of previous results and how this work fits into the field are a necessity. Even then, with the current results the manuscript would not be on the level of a Nature Communications article and should be submitted to a more specialized journal.

Minor: Figure 2c caption says “black double arrow” when the arrow is gray.

Thank you for pointing out the typo. We have corrected it in the revised manuscript.

References:

Ritzkowsky, F. et al. Tailoring the impulse response of petahertz optical field-sampling devices. Int. Conf. Ultrafast Phenom. (UP), Optica Tech. Digest, Th1A.4 (2022).
Zimin, D. et al. Petahertz-scale nonlinear photoconductive sampling in air. Optica 8(5), 586-590 (2021).

We thank the reviewer for recognizing the quality of our work. We hope that our detailed answers and additional experiments shown in Fig.R2 and Fig.R3 will convince the reviewer about the quality of our work.

REVIEWER COMMENTS

Reviewer #1 (Remarks to the Author):

In the revised version of the manuscript, the authors have successfully addressed the comments by myself and the other referees. Furthermore they have also replied to the main criticism by Referee 4, which is the incremental novelty with respect to the papers by Yang et al. and by Zimin et al. I agree with the authors that the paper by Zimin et al is not comparable to their study, while the Yang paper uses a similar nanodevice configuration but does not track coherent electron oscillations using homodyne beating. I am therefore in favour of publication of the current version of the paper in Nature Communications.

Reviewer #2 (Remarks to the Author):

The authors have provided a somewhat lengthy response to the questions of the Reviewers. The questions that I have raised in my first report have been addressed satisfactorily.

I am not certain, however, that also the pertinent questions of Reviewers 3 and 4 have fully been addressed.

Reviewer #3 (Remarks to the Author):

In my first report I indicated that the major problem of this manuscript is an absence of any solid proof in support of the gap size considered by the authors to explain their data. The amended version of the manuscript fails to convincingly address this point. Therefore, I consider that the manuscript is not suitable for the publication in Nature Communications.

Honestly, it is not suitable for the publication in any scientific journal because one of the main requirements to the scientific work is its reproductivity, which is absolutely not the case for the present contribution.

1. The authors discuss that antennas with different gap sizes of 5, 0, 5, 10 nm are fabricated. However, the information is not given on what is the structure (gap size targeted during nanofabrication) actually used to measure current reported in figures 2, 3, and 4. Furthermore, starting from the gap size of 10 nm or even 5 nm and arriving to 1.2 nm used in the calculation (such a precision!) is a big change. The authors speculate that this can be owing to the atom migration. The same migration for all antennas? If the migration is caused by the tunneling currents how to be sure that part of the intensity dependence is owing to this very effect and not to the physics discussed by the authors? Where are the figures showing reproductivity of the data? Where are the figures showing results for different geometries of the fabricated antennas?

2. What is the gap size used for the electromagnetic calculations shown in figure 2. Is it 10 nm as one would guess from the only results discussed in the paper and SI? But then, it contradicts to the assumption of the 1.2 nm gap performed in the transport calculations. Where are the results of the electromagnetic calculations performed for the gap size of 1.2 nm? Notice that for the 1.2 nm gap the frequency of the plasmon resonance, and the field enhancement (and thus the estimates for the Keldysh

parameter) should be very different from that calculated for the 10 nm gap.

3. I attract the attention of the Authors to the recent work: "Continuous-wave multiphoton-induced electron transfer in tunnel junctions driven by intense plasmonic fields" ACS Photonics doi: 10.1021/acsp Photonics.3c00714. It has much in common with present contribution, but has a merit of the adequate description of the experimental procedures and theoretical models. It also operates with much more realistic model of the tunneling barrier than that used in the present work.

Reviewer #4 (Remarks to the Author):

I want to thank the authors for the additional modeling and the new device measurements. These additions help bolster their claims of novelty for the optical homodyne technique and plasmon oscillation model.

I welcome the explanation in their rebuttal letter about the importance of this incremental step forward toward understanding laser-induced electron oscillations. Although the technique appears to be identical to the previous work of Yang (now source 42), except for the minor difference as mentioned before of an AOFS, I can appreciate that the modeling and discussion focus on the understanding of the plasmon oscillations instead of just a direct map from photocurrent onto CEP. I still find it inappropriate that this nearly-identical work is only mentioned for its minor section on dielectric breakdown, instead of as a starting point to compare the models earlier in the manuscript.

While the plasmon oscillations in the bowtie nanojunction is an interesting framework toward better modeling the laser electric field, and I believe this is a well-written and displayed manuscript, I still fail to see why this is an important enough step forward that justifies publication in Nature Communications. I support its publication, but suggest a more specialized journal such as Communications Physics.

Response to the referees

We would like to cordially thank all the referees for their invaluable comments. We sincerely thank reviewer #1 and reviewer #2 for recommending the publication of our work in Nature Communications. Below we enlist detailed responses to the comments of reviewer #3 and reviewer #4. We have revised the manuscript in accordance with their suggestions and criticisms. We have marked essential revisions as **REV#** and highlighted the revised text in the main-text and supplementary materials (SM) to ease the evaluation.

Reviewer #3 (Remarks to the Author):

Reviewer report on the manuscript by Y. Luo and co-authors entitled "Real time tracking of coherent oscillations of electrons in a nanodevice by photo-assisted tunneling".

In my first report I indicated that the major problem of this manuscript is an absence of any solid proof in support of the gap size considered by the authors to explain their data. The amended version of the manuscript fails to convincingly address this point. Therefore, I consider that the manuscript is not suitable for the publication in Nature Communications.

Honestly, it is not suitable for the publication in any scientific journal because one of the main requirements to the scientific work is its reproductivity, which is absolutely not the case for the present contribution.

We thank the reviewer for his/her time and efforts in the review process. We have made substantial efforts to address the reviewer's concerns, and we respectfully disagree with his/her comments regarding the validity of our findings.

We understand the importance of gap sizes in experiments exploring plasmonic response in bowtie nanoantennas. However, our purpose in the current work is not to precisely control and measure the ~ nm gap sizes, primarily owing to inherent limitations in the nanofabrication processes. Those limitations include the (uncontrollable) deviation between designed and fabricated junction gap sizes, and the limited spatial resolution of the SEM. These challenges are common in experiments involving bowtie structures, particularly in the few-nm region, not specific to our work.

As transparently stated in the manuscript, the primary focus of our work is on the real-time tracking and phase control of the collective electron oscillations in a plasmonic nanodevice, as well as the precise determination of the contributions of linear and nonlinear electron oscillations in the generated electron oscillations. We had performed measurements of the photocurrent in the device as a function of the applied bias (I-V curve) and explained the nonlinear behavior of the photocurrent using a 1-D tunneling model. The intensity dependence measurements, bias dependence measurements and phase control measurements were performed with high signal stability, and the results are fully reproducible.

The inability to precisely control the local structure (at atomic scales) of the nanodevices in the nanofabrication process does not compromise the key findings of our study, which focuses on characterizing the plasmonic response directly in the time domain for the very first time. In the following, we provide detailed response to the reviewer's comments.

1. The authors discuss that antennas with different gap sizes of 5, 0, 5, 10 nm are fabricated. However, the information is not given on what is the structure (gap size targeted during nanofabrication) actually used to measure current reported in figures 2, 3, and 4. ...

We thank the reviewer for pointing out that the designed gap size in the nanodevices was not mentioned in the manuscript. All the measurements presented in the main-text (data for Fig. 2, Fig.3 and Fig. 4) were conducted on the same nanodevice with a targeted gap size of 5 nm. This information is now included in the caption of the Fig. 2 (main-text), please see **REV#1: 'The designed gap size of the nanoantennas was 5 nm (see SM, section I)'**.

...Furthermore, starting from the gap size of 10 nm or even 5 nm and arriving to 1.2 nm used in the calculation (such a precision!) is a big change. The authors speculate that this can be owing to the atom migration. The same migration for all antennas? ...

As discussed in Section 1.1 of the SM, the fabricated gap sizes will differ from the designed values due to the proximity effect and the exposure characteristics of the resist, which is a common challenge in the nanofabrication processes. **Obtaining an actual gap size of ~ 1 nm from a designed gap size of 5 or 10 nm is well within the expected range, considering the standard deviation in the nanofabrication of ~ 5 nm as shown previously by Kaniber *et al.* Scientific Reports 6, 23203 (2016) and ~3-10 nm by Yang *et al.* Nature Communications 11, 3407 (2020).** In the previous review round, we showed different time traces of the laser-induced photocurrent for two nanodevices with the same designed junction gap size of 10 nm. The measured photocurrent as a function of the delay between two laser pulses showed clear oscillations for the first nanodevice (A-1), whereas the oscillations for the second nanodevice (A-2) were not clear. This difference is attributed to the difficulty of achieving precise control over the junction gap sizes and the atomic-scale local structures at the apexes of the fabricated nanoantennas.

The nonlinear behavior of the laser-induced current in both the intensity dependence (Fig. 3e) and the bias dependence (Fig. 4a) measurements imply that the underlying mechanism of current generation in the nanodevices is dominated by photon-assisted tunneling. As also agreed by the reviewer, '*50 Angstrom gap is way too large to talk about tunneling*', a gap size of ~1 nm is therefore be considered for understanding the experimental observations. Our modeling of the I-V curve in the tunneling regime assumes an effective tunneling gap of 1.2 nm, providing a reasonable explanation for our experimental observations. **However, we want to emphasize that this value is a fitting parameter and our simulation is not intended for precise determination of the actual tunneling gap size, given the simplified nature of the 1-D tunneling model.** The actual gap sizes of the 7 antennas may differ from each other.

In order to overcome this limitation in the measurements, we had performed control experiments to validate our findings in the previous review round. We had fabricated nanodevices comprising single nanoantennas instead of arrays of nanoantennas. **Single nanoantenna devices have the advantage of featuring only a single gap size instead of an effective tunneling gap size.** The results of these measurements are shown in Fig. S5 of the methods section.

Electromigration of Au atoms in the nanoantenna junction can further modify the junctions, as discussed by Yang *et al.* Nature Communications 11, 3407 (2020). To avoid misunderstanding, we have modified the sentence in the main-text discussing the uncertainty of the actual gap sizes,

please see **REV#2**: *'The deviations between the designed and the actual junction gap sizes of the nanodevice can result from exposure characteristics of the resist during the nanofabrications, or the electromigration of Au atoms in the nanoantenna junction.'*

...If the migration is caused by the tunneling currents how to be sure that part of the intensity dependence is owing to this very effect and not to the physics discussed by the authors? Where are the figures showing reproducibility of the data?

We would like to emphasize that below the damage threshold intensity of the incident laser pulses on the nanodevices, we observe no discernible changes in the laser-induced photocurrent, suggesting no changes at the apexes of the nanoantennas. The nanodevice was stable during the intensity dependence measurements (Fig. 3e of the main-text) using pulse energies below the damage threshold.

To confirm the reproducibility of our data, we had recorded the time traces of the photocurrent **before starting** the intensity dependence measurement (Fig. 3e of the main-text) and **after finishing** the measurements, as shown in Fig. R1. The magnitudes of the linear and nonlinear photocurrents at zero-delay time in Fig. R1 are identical to the photocurrents shown in Fig. 3e. This observation corroborates the conclusion that the junction gaps and the local structure in the nanoantennas remained unaffected during the intensity dependence measurements.

We have now included the reproducibility check measurements in the Supplementary Information (Fig. S6). We have also incorporated a sentence in the main-text to address concerns about atom migration and damage, please see **REV#3**: *'The local structures in the nanoantennas are unaffected during the intensity dependence measurement (Fig. 3e) as confirmed by the reproducibility-check measurements (see Fig. S6 in SM).'*

Fig. R1 | Reproducibility of the intensity dependence measurements. **a, b**, Variation of the laser-induced photocurrent as a function of the delay between pulse-1 and pulse-2 laser pulses measured at the lock-in frequency of f_0 (**a**) and $2f_0$ (**b**) **before starting** the intensity dependence measurements shown in Fig. 3 of the main-text. **c, d**, Variation of the laser-induced photocurrent as a function of the delay between pulse-1 and pulse-2 laser pulses measured at the lock-in frequency of f_0 (**c**) and $2f_0$ (**d**) **after finishing** the intensity dependence measurements shown in Fig. 3 of the main-text. The pulse energy of the laser pulses was set at ~ 200 pJ and the bias in the nanoantenna junction was 3.0 V.

...Where are the figures showing results for different geometries of the fabricated antennas?

We did observe variations in photocurrents from different nanodevices, indicating differences in their plasmonic responses, which are possibly due to the different nanostructures. In response to the reviewer's current query, we have further included the measurements on another nanodevice with a different plasmonic response, as shown below in Fig. R2. The variations in both the time and spectral domain responses for linear and nonlinear electron oscillations are evident. We would like to highlight that the variation of plasmonic response is possibly related to the different local structures at the apexes of the nanoantennas. However, a systematic study of the variation of the local plasmonic and nonlinear electron oscillations for different device geometries is beyond the focus of the current work.

We have now included the results for different geometries of the fabricated antennas in the Supplementary Information (Fig. S7), please see **REV#4**.

Fig. R2 | Plasmon and nonlinear electron oscillations for different nanodevice geometry.

a, b, The variation of the linear (**a**) and nonlinear (**b**) laser-induced photocurrents measured as a function of the delay between pulse-1 and pulse-2 pulses for the nanodevice 'A-3' designed with a targeted gap size of -5 nm. The pulse energy of the laser pulses was set at ~ 200 pJ and the bias in the nanoantenna junction was 3.0 V. **c, d,** The temporal variation of the linear (**c**) and nonlinear (**d**) laser-induced photocurrents for a nanodevice 'A-0' (reproduced from Fig. 2a and Fig. 3a of the main-text), respectively. The nanoantenna for the measurements shown in the main-text is labeled as A-0. **e, f,** Comparison of the measured spectral response of the linear (**c**) and nonlinear (**d**) time-resolved electron oscillations between nanodevices 'A-3' and 'A-0'.

2. What is the gap size used for the electromagnetic calculations shown in figure 2. Is it 10 nm as one would guess from the only results discussed in the paper and SI? But then, it contradicts to the assumption of the 1.2 nm gap performed in the transport calculations. Where are the results of the electromagnetic calculations performed for the gap size of 1.2 nm? Notice that for the 1.2 nm gap the frequency of the plasmon resonance, and the field enhancement (and thus the estimates for the Keldysh parameter) should be very different from that calculated for the 10 nm gap.

The finite element simulations for estimating the field enhancement in the nanoantenna junction have been performed with a gap size of 10 nm. The simulated plasmonic spectrum is in qualitative agreement with our experimental observations, providing valuable insights into the overall plasmonic response.

To the best of our understanding, while the local field enhancement factors indicated experimentally may be reliable, factors derived from the electromagnetic simulations have limitations. The key evidence to identify the light-matter interaction regime is the intensity dependence curve, as demonstrated in Fig. 3 of the main-text. The observed nonlinear intensity dependence of the photocurrents in the nanodevices indicates that the interaction regime is perturbative (Keldysh parameter > 1) and not nonperturbative (Keldysh parameter < 1).

We understand that plasmonic properties, such as the frequency of plasmon resonance and field enhancement, can vary significantly with gap size, particularly in the transition from 10 nm to 1.2 nm. We also recognize the impact of the local structure of the apexes on the plasmonic response, which remains uncertain and can contribute to variations as observed in Figure R2. We appreciate the suggestion to explore simulations within the 1-nm region, where quantum charge-transfer effects may become relevant. However, we propose considering those simulations in future studies, as they extend beyond the specific focus of the current work. We hope for the understanding of the reviewer.

3. I attract the attention of the Authors to the recent work: “Continuous-wave multiphoton-induced electron transfer in tunnel junctions driven by intense plasmonic fields” ACS Photonics doi: 10.1021/acsp Photonics.3c00714. It has much in common with present contribution, but has a merit of the adequate description of the experimental procedures and theoretical models. It also operates with much more realistic model of the tunneling barrier than that used in the present work.

We appreciate the reference of the reviewer on the ACS Photonics work. While we acknowledge the relevance of the mentioned work in investigating photon-induced tunneling currents in plasmonic junctions, we want to highlight the significant differences between this work and our studies. Our work focuses on the real-time tracking and phase control of collective electron oscillations in a plasmonic nanodevice, as well as the precise determination of the contributions from linear and nonlinear electron oscillations in the generated tunnelling currents. This is distinct from the ACS Photonics paper, which primarily deals with continuous-wave photon-induced electron transfer in the tunnel junction of an STM.

Regarding the theoretical model, we acknowledge the detailed descriptions in the ACS Photonics work. However, it is important to note that there are no fundamental differences between their tunneling model and the model considered in our work. Both studies begin with the one-dimensional potential barrier model, calculate the potential barrier with the image potential considered (following J. G. Simmon, J. Appl. Phys. 34, 1793–1803 (1963)), and then numerically calculate the tunneling probability. We really cannot find their model to be more “realistic”. We have avoided making an extensive discussion on the employed theoretical models as this has been thoroughly described in many earlier works (such as Ref. #45 and #46).

Reviewer #4 (Remarks to the Author):

I want to thank the authors for the additional modeling and the new device measurements. These additions help bolster their claims of novelty for the optical homodyne technique and plasmon oscillation model.

We thank the reviewer for his/her appreciation of our work.

I welcome the explanation in their rebuttal letter about the importance of this incremental step forward toward understanding laser-induced electron oscillations. Although the technique appears to be identical to the previous work of Yang (now source 42), except for the minor difference as mentioned before of an AOFS, I can appreciate that the modeling and discussion focus on the understanding of the plasmon oscillations instead of just a direct map from photocurrent onto CEP. I still find it inappropriate that this nearly-identical work is only mentioned for its minor section on dielectric breakdown, instead of as a starting point to compare the models earlier in the manuscript.

We appreciate the reviewer's recognition of our efforts in advancing the understanding of laser-induced electron oscillations. We understand the concern about the appropriate mention of this related work by Yang *et al.* Nature Communications 11, 3407 (2020), and we value the insights provided by Yang *et al.*'s study.

We have modified the introductory part of our manuscript to highlight the relevance and value of Yang *et al.*'s work in the context of on-chip ultrafast science. Please see **REV#5** in the main-text: *'Recent experiments have shown that the photo-assisted tunnelling currents induced in a plasmonic nanodevice can be controlled by tuning the carrier-envelope-phase (CEP) of the driving laser pulses.'*

Please note that the **Ref. #42** has been renumbered to **#19** in the revised manuscript.

While the plasmon oscillations in the bowtie nanojunction is an interesting framework toward better modeling the laser electric field, and I believe this is a well-written and displayed manuscript, I still fail to see why this is an important enough step forward that justifies publication in Nature Communications. I support its publication, but suggest a more specialized journal such as Communications Physics.

We once again thank the reviewer for his/her positive instance of our work. We hope that the revisions in the manuscript and the quality of our work will now convince the reviewer and he/she will now recommend the work for publication in Nature Communications.

REVIEWERS' COMMENTS

Reviewer #3 (Remarks to the Author):

I am now satisfied with author's reply on the issues raised in my reports and in reports by other reviewers. There is only one point left which goes along my statement of the necessity to clearly present what and how is done:

If the Authors will state in the caption of Fig.2 that the electromagnetic calculations were performed for the gap size of 10 nm, I will support the publication of this paper in Nature Communications.